# A Cadaveric Study Using Computed Tomography for Measuring the Ocular Bulb and Scleral Skeleton of the Atlantic Puffin (Aves, *Alcidae*, *Fratercula arctica*)

**DOI:** 10.3390/ani13152418

**Published:** 2023-07-26

**Authors:** Marcos Fumero-Hernández, Mario Encinoso, Ana Sofia Ramírez, Inmaculada Morales, Alejandro Suárez Pérez, José Raduan Jaber

**Affiliations:** 1Veterinary Hospital, Faculty of Veterinary Medicine, University of Las Palmas de Gran Canaria, Trasmontaña, 35413 Arucas, Las Palmas, Spain; marcos.fumero101@alu.ulpgc.es (M.F.-H.); inmaculada.morales@ulpgc.es (I.M.); 2Department of Pathology and Food Technology, Faculty of Veterinary Medicine, Universidad de Las Palmas de Gran Canaria, Trasmontaña, 35413 Arucas, Las Palmas, Spain; anasofia.ramirez@ulpgc.es (A.S.R.); alejandro.suarezperez@ulpgc.es (A.S.P.); 3Department of Morphology, Faculty of Veterinary Medicine, University of Las Palmas de Gran Canaria, Trasmontaña, 35413 Arucas, Las Palmas, Spain

**Keywords:** computed tomography, imaging, ocular morphometry, orbit, sclerotic ring, birds

## Abstract

**Simple Summary:**

Advanced diagnostic imaging techniques, such as CT, can provide helpful information on the specific structures of the head, such as the ocular bulb, due to their high spatial resolution, avoidance of overlapping structures, and fast imaging acquisition. An adequate knowledge of the eye bird anatomy is essential for clinicians, biologists, and researchers to understand many aspects concerning its biology.

**Abstract:**

Imaging diagnosis plays a fundamental role in avian medicine. However, there are few publications regarding its use in ophthalmology. Seabirds, in particular, present a peculiar ecology since their lives take place in very diverse environments: the aquatic, the terrestrial, and the aerial. This fact implies a series of adaptations at a visual level that are necessary for adequate interaction with the environment. Therefore, knowledge of eye particularities is of great importance for the scientific community since it allows us to deepen our understanding of the ocular anatomy and biology of these animals, which are increasingly present in veterinary and wildlife centers. In our study, we performed a morphometric analysis of the ocular bulb and its internal structures in the puffin *(Fratercula arctica*) using advanced imaging techniques such as CT.

## 1. Introduction

The Atlantic puffin (*Fratercula arctica*) is a member of the Alcidae family and one of three species in the Fratercula genus, alongside the horned puffin (*Fratercula corniculata*) and the tufted puffin (*Fratercula cirrhata*), which are typically found in the North Pacific [1]. These medium-sized alcids have a wingspan of approximately 50 cm and can weigh up to half a kilogram. They possess dense black plumage on their head, neck, and dorsum, with white patches on their chest, ventrum, and around the eyes. The puffin’s bill is large and exhibits orange tones, which fade to less intense colors after the breeding season. Notably, there is no distinct sexual dimorphism, although males are generally slightly larger than females and their colors become more conspicuous during the mating season [2,3,4,5,6].

The Atlantic puffin has a wide distribution, inhabiting the entire northern Atlantic Ocean region, from northwestern Greenland, Newfoundland, and Maine (USA) in the west to northwestern Russia and the Canary Islands (Spain) in the east [1,7]. These birds spend the majority of their lives in the ocean, returning to land solely for reproduction. They primarily nest on isolated islands and cliff sides, ranging from Brittany and the Bay of Fundy (Canada) to the Arctic sea ice on both sides of the Atlantic. Puffins return to their colonies in March to nest during April and May and then leave in July to return to the open sea [5,8,9]. They are monogamous birds, typically selecting the same partner and nesting location in successive breeding periods. Upon returning to the breeding area, males primarily engage in nest construction, which involves shallow burrows, rocky ledges, and crevices, sometimes, incorporating plant material [5,10,11]. Females, after laying a single egg per season, dedicate themselves to incubating it and subsequently feeding the chick [12,13,14]. Chicks are primarily fed with fish, particularly herring (*Clupea harengus*; 12–85%), hake (*Urophycis* spp., *Merluccius* spp.; 12–87%), or sand eel (*Ammodytes* spp.; 0–71%). Adult puffins are also piscivorous, with varying amounts of polychaetes and crustaceans in their diet [5,14,15].

The Atlantic puffin is currently listed as a threatened species on the International Union for Conservation of Nature (IUCN) Red List, categorized as vulnerable, due to a significant population decline over most of its range [1]. Climate change, linked to ocean temperature variations that impact plankton blooms and disrupt the food chain, is one of the main factors contributing to their vulnerability [16,17]. Puffins are also sensitive to extreme weather events, such as marine storms, which have been increasing in frequency and intensity and are associated with mass mortality [18,19]. Human activities, including hunting for consumption and oil spills or other pollutants, also negatively impact puffin populations [20,21]. Therefore, the conservation of puffins and other seabirds is a globally significant task. Efforts aimed at their conservation require a deeper understanding of their anatomy and biology as they relate to their behavior and the threats they face.

Numerous studies have emphasized the importance of vision in the interaction of these animals with their environment and their overall survival [22,23,24]. Bird vision ranks among the sharpest in the animal kingdom. Avian eyes possess unique anatomical features, such as the pecten and sclerotic ring, found only in birds and certain reptiles. These adaptations enable birds to effectively interact and navigate across diverse environments, encompassing aquatic, terrestrial, and aerial realms [25,26,27]. Consequently, comprehending the ocular structures and functionality of the Atlantic puffin holds significant importance for veterinarians, biologists, and the wider scientific community.

Studying internal animal anatomy can be highly intricate. While dissection has traditionally been the conventional approach, contemporary diagnostic imaging techniques such as magnetic resonance imaging (MRI) and computed tomography (CT) offer minimally invasive and efficient means to obtain precise anatomical information [28]. Although primarily employed in human medicine, smaller versions of these imaging modalities, such as micro-computed tomography, have been developed to minimize image distortion, proving helpful for investigating the anatomy of small animals [29,30,31].

In the case of puffins, the existing literature predominantly focuses on various aspects of their biology and ecology [9,14,21,32] or approaches ophthalmology from a clinical standpoint [33]. However, conducting anatomical and morphobiometric analyses of the puffin’s eye and associated structures can yield valuable insights into animal behavior patterns. This has been demonstrated in studies exploring the fossils of extinct marine reptiles belonging to the *Ichthyosaur* order [34] and non-passerine birds [35]. Similar investigations have also been conducted on live animals, encompassing both domestic and wild species [36,37,38].

Therefore, the objective of this study was to employ non-invasive examinations, such as computed tomography, to measure the size and characteristics of the puffin’s eye and its associated structures.

## 2. Materials and Methods

### 2.1. Animals

In order to conduct this study, a total of 29 Atlantic puffin carcasses were obtained by the Consejeria de Área de Medio Ambiente, Clima, Energía y Conocimiento of the Cabildo de Gran Canaria (Gran Canaria, Canary Islands, Spain). The weight of the animals ranged from 0.185 kg to 0.251 kg, with a mean weight of 0.251 kg. Since the physical examination alone did not provide sufficient information to determine the precise cause of the stranded individuals, modern non-invasive imaging techniques were employed to assess their appendicular skeleton, identify potential metabolic bone diseases, and exclude the presence of foreign bodies and internal organ injuries. Prior consent was obtained from the responsible person at the Cabildo de Gran Canaria to include the puffins in this study. The scanning procedures were carried out at the Veterinary Hospital of Las Palmas de Gran Canaria University.

### 2.2. CT Technique

CT examinations of the skull were conducted on the thawed carcasses of these specimens after a 12-h defrosting period at room temperature. Sequential slices were acquired using a 16-slice helical CT scanner (Toshiba Astelion, Canon Medical System^®^, Tokyo, Japan). The animals were positioned symmetrically in a prone position on the stretcher, employing a craniocaudal entry. A standard clinical protocol was employed, utilizing parameters of 120 kVp, 80 mA, a 512 × 512 acquisition matrix, an 1809 × 858 field of view, a pitch of 0.94, and a gantry rotation of 1.5 s. The resulting images had a thickness of 0.6 mm. CT scans were obtained in the dorsal, transverse, and sagittal planes with both bone and soft tissue windows. Subsequently, all the acquired images were imported into an image viewer (OsiriX MD, Apple, Cupertino, CA, USA) to facilitate data manipulation and measurements of the puffin ocular bulb and its associated structures.

### 2.3. Measurements

We conducted measurements on the head length, width, and orbit depth. Additionally, measurements of both eyes (*n* = 58) were analyzed by two observers using oblique sagittal, transverse, and dorsal CT images obtained from all the skulls. A soft tissue attenuation window was utilized for image analysis. The measurement methodology followed a protocol previously described in studies on the loggerhead turtle [39], as well as in dogs and cats [36,37,38], with some modifications based on studies conducted on non-passerine birds [35]. The parameters and specific measurements taken are described as follows:(A)Measurements in the transverse plane relative to the ocular bulb:
-Lens diameter: This parameter refers to the maximum distance between the lateral and medial edges of the lens, also known as the equatorial diameter (Figure 1A);-Internal diameter of the sclerotic ring, which represents the maximum distance between the inner lateromedial edges of the ring close to the cornea (Figure 1A);-External diameter of the sclerotic ring, which corresponds to the maximum distance between the outer lateromedial edges of the ring close to the sclera (Figure 1A);-Thickness of the sclerotic ring, defined as the distance between the internal diameter of the sclerotic ring and the external diameter, measured in the dorsal arch (Figure 2).(B)Transverse plane concerning the puffin’s body


Height of the ocular bulb, which corresponds to the distance between the dorsal part of the *os frontale* and the *os quadratojugale*, indicating the vertical dimension of the ocular bulb (Figure 3B).
-Attenuation of the sclerotic ring and lens: This measurement was taken in the dorsal area and expressed in Hounsfield Units, providing information about the radiodensity of the sclerotic ring, lens, and vitreous humor (Figure 4).
(C)Dorsal plane
-Width of the ocular bulb: This measurement represents the lateromedial length of the ocular bulb, extending from the *os lacrimale* to the inner face of the *os frontale* (Figure 3A);-Length of the ocular bulb: This measurement refers to the distance between the most rostral and caudal portions of the ocular bulb, spanning from the *septum interorbitale* to the *processus antorbitalis* (Figure 3A).
(D)Three-dimensional reconstruction
-Head length: This measurement is obtained from the centre point of the junction between the beak joint to the *prominentia cerebellaris*, providing the length of the head (Figure 5A);-Depth of the orbit: This measurement represents the distance between the midpoint of the orbital diameter and the *foramen opticum*, indicating the depth of the orbit (Figure 5B).


### 2.4. Statistical Analysis

The statistical analysis was performed using commercially available software (SPSS 19, Statistical Package for the Social Sciences, Chicago, IL, USA). Descriptive statistics, including the mean, median, range, and standard deviation (SD), were calculated for each measurement. The Shapiro-Wilk test was used to assess the normal distribution of quantitative data. The Mann-Whitney U test was employed to compare measurements between the right and left eyes. Furthermore, Spearman correlation was utilized to analyze the relationship between different ocular bulb variables and head length, as well as other quantitative variables. The statistical significance level was set at *p* < 0.05.

## 3. Results

In the computed tomography images, the sclerotic rings are visible from the anterior view of the respective eyes as circular structures with continuous morphology, distinct from the surrounding elements of the skull. The ellipsoid-shaped ocular bulbs exhibit well-defined soft tissue margins that likely correspond to the sclera. Various components of the eye, including the lens, vitreous and aqueous humors, and the anterior and posterior chambers, are also well distinguished in the images (Figure 1, Figure 2, Figure 3, Figure 4 and Figure 5). From these images, we acquired the different measurements corresponding to both eyes (Table 1).

Table 1 presents the measurements of internal structures for the left, right, and both eyes of all 29 puffins included in the study. The mean, median, and standard deviation of the head length were 5.10 cm, 5.07 cm, and 0.13 cm, respectively, with a range of 4.8 to 5.32 cm. The average lens diameter for all eyes was 3.32 mm, with a range of 2.7 to 4.2 mm. The average internal diameter of the sclerotic ring was 0.63 cm, ranging from 0.59 to 0.7 cm. Additionally, the average external diameter of the sclerotic ring was 1.44 cm, ranging from 1.38 to 1.49 cm. The sclerotic ring thickness had a mean value of 2.16 mm, with a range of 2.04 to 2.29 mm. Regarding the ocular bulbs, they had an average height of 1.53 cm (range: 1.41–1.61 cm), an average width of 1.28 cm (range: 1.1–1.42 cm), and an average length of 1.61 cm (range: 1.46–1.71 cm). Additionally, the orbit depth had a mean of 8.19 mm (range: 7.9–8.5). Some variables exhibited a non-normal distribution. The Mann–Whitney U test indicated no significant differences between the measurements of the right and left eyes. Likewise, the same analysis demonstrated no statistically significant differences between the right and left eyes when considering all variables collectively.

The Spearman correlation analysis conducted to examine the relationship between eye measurements and head length did not reveal any statistically significant findings. However, significant correlations were observed in other aspects. There was a strong correlation between ocular bulb height and width (rho = 0.661; *p* < 0.001), indicating a substantial relationship between these variables. Additionally, moderate correlations were observed between ocular bulb height and length (rho = 0.403; *p* = 0.002), as well as between ocular bulb width and length (rho = 0.449; *p* = 0.002), suggesting meaningful associations between these measurements. Moreover, a good correlation was found between the external and internal diameters of the sclerotic ring (rho = 0.431; *p* < 0.001), indicating a relationship between these variables.

Correlations involving the internal diameter of the sclerotic ring showed statistical significance with ocular bulb height (rho = 0.601; *p* < 0.001), length (rho = 0.496; *p* < 0.001), but not with width (rho = 0.203; *p* = 0.127). In a similar way, the external diameter of the sclerotic ring exhibited significant correlations with ocular bulb height (rho = 0.688; *p* < 0.001) and width (rho = 0.642; *p* < 0.001), while in this case no significant correlation was found with ocular bulb length (rho = −0.059; *p* = 0.662). Significant correlations were observed when comparing the orbit depth with ocular bulb measurements: height (rho = −0.484; *p* < 0.001) and width (rho = −0.294; *p* = 0.025). However, no significant correlation was observed with ocular bulb length (rho = −0.114; *p* = 0.395).

In terms of attenuation, the regions of interest (ROI) for measuring the attenuations of the sclerotic ring, vitreous humor, and lens were indicated by the green, yellow, and purple circles, respectively (Figure 4). It is noteworthy that the vitreous chamber exhibited hypoattenuation compared to the sclerotic ring. Conversely, the lens displayed hyperattenuation in relation to the surrounding aqueous humor. The mean lens attenuation was determined to be 69.53 Hounsfield units (HU), with a range of 64–79 HU. The vitreous humor exhibited an average attenuation of 35.38 HU (range: 31–45 HU), while the sclerotic ring displayed an attenuation of 720.50 HU (range: 687–752 HU).

## 4. Discussion

In this study, we conducted computed tomography to assess the ocular bulb and associated hard tissues of the Atlantic puffin. To the best of the authors’ knowledge, this is the first report describing measurements of Atlantic puffin eyes using a CT scan. This technique has proven quite useful in the evaluation of the morphological knowledge of the normal eye and associated structures in different species, such as dogs [36,37], cats [38], and reptiles [39,40]. CT facilitates the view of body sections from different tomographic planes, offering images with high anatomic resolution without tissue overlapping, adequate contrast between different structures, and excellent tissue differentiation [41,42]. Consequently, modern imaging techniques like CT have significantly improved the image quality, enabling exceptional evaluation of eye dimensions and different structures of the puffin ocular bulb, including the sclerotic ring, vitreous chamber, and lens. This anatomical knowledge is crucial for understanding the biology and ophthalmology of seabird species. Thus, the tissue structures related to the ocular bulb, including the sclerotic ring and the orbit, play a vital role in inferring the size and shape of the ocular bulb, which have been linked to the bird’s activity patterns [35]. It is crucial to emphasize that mammals lack a sclerotic ring, and therefore, the sole bone correlation lies in the morphology of the orbit itself [35]. Nevertheless, studies on primates have indicated that as body size increases, the volume of the primate orbit increases at a higher rate compared to the volume of the eye. Consequently, in larger bodies, the dimensions of the orbit may not accurately predict the size or shape of the eye [43,44,45,46]. Our study does not compare body size with eye measurements. Instead, we focused on the correlation between head length and eye measurements, as previous studies on birds have established this association [35]. However, we did not observe any significant correlation between the eye measurement and head length in Atlantic puffins.

The morphometric analysis of the internal diameter of the sclerotic ring provides valuable insights into the relative size of the dilated pupil and the cornea, as this bone correlates well with the cornea. This relationship enables the estimation of the light-capturing capacity of the eye and consequently deduces the behavior pattern, whether diurnal or nocturnal [25,35]. However, it is important to note that this correlation may not be reliable if analyzed in isolation. In bird species that do not exhibit distinct activity patterns strongly skewed towards day or night, it becomes necessary to combine these measurements with those of the orbit depth to obtain more accurate and reliable results because just the dimensions of the sclerotic ring alone have not separated nocturnal and diurnal birds well [35].

Our CT studies have also revealed a significant and positive statistical correlation between the internal and external diameters of the sclerotic ring and the width, height, and length of the ocular bulb. These findings suggest a potential relationship between these parameters, visual acuity, and the diurnal pattern observed in the Atlantic puffin. Similar associations have been documented in extinct and extant bird species [35] and sea turtles [39]. However, further investigations focusing on the influence of water pressure during diving or air pressure on the Atlantic puffin’s ocular bulb sclerotic ring should be performed to deepen our understanding in this area.

The assessment of orbit depth, from the midpoint of the orbital diameter to the optic foramen, has been previously investigated by other researchers in extinct and extant birds [35]. In those studies, the measurements were performed directly on the carcasses rather than indirectly, as in our study, where three-dimensional reconstruction techniques and specialized computer programs were employed. In our animals, significant correlations were observed when comparing the orbit depth with ocular bulb height and width, suggesting relevant animal visual acuity and sensitivity to low light. Close findings were observed in extinct aquatic reptiles such as *Ichthyosaurus* and *Ophtalmosaurus* with large eyes and similar feeding habits to Atlantic puffins [34], which spend part of the day feeding on small, fast-moving preys at depths ranging from 30–60 m [1,2]. However, it is essential to acknowledge that these measurements may vary and could be influenced by operator-dependent factors during the contouring of CT images, as previously reported in other studies [36,38].

The quality of the acquired tomographic images presented here could be influenced primarily by two factors: the utilization of conventional CT instead of micro-CT and the small size of the birds included in the study. Previous studies have demonstrated that micro-CT offers high contrast and superior image quality for small animals and tissues [47]. However, it is important to highlight that, despite using a conventional CT scanner, we obtained the essential information required to fulfill the study objectives.

## 5. Conclusions

In this study, the CT images obtained in different planes provided relevant information about the morphometric characteristics of the ocular bulb and the sclerotic ring in the Atlantic puffin. The reference values established included the presumed normal diameters of the ocular bulb and lens, as well as various measurements of the sclerotic ring, which may be associated with the visual capabilities and activity patterns of these seabird species. However, it is essential to note that further investigations involving live animals are necessary to evaluate potential differences compared to data obtained from carcasses. Additionally, it is essential to consider the inherent operational error associated with the manual contouring of computed tomography images.

## Figures and Tables

**Figure 1 animals-13-02418-f001:**
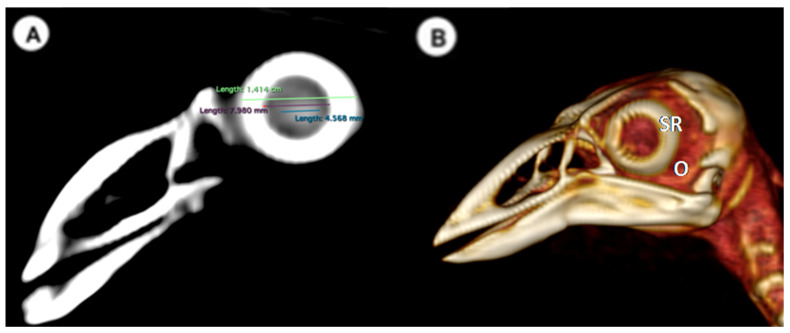
(**A**) Transverse multiplanar reconstruction (MPR) related to the ocular bulb of *F. Arctica* with measurements of the lens and the internal and external diameters of the sclerotic ring. (**B**) Volume rendering image of the *F. Arctica* skull with the sclerotic ring (SR) and orbit (O).

**Figure 2 animals-13-02418-f002:**
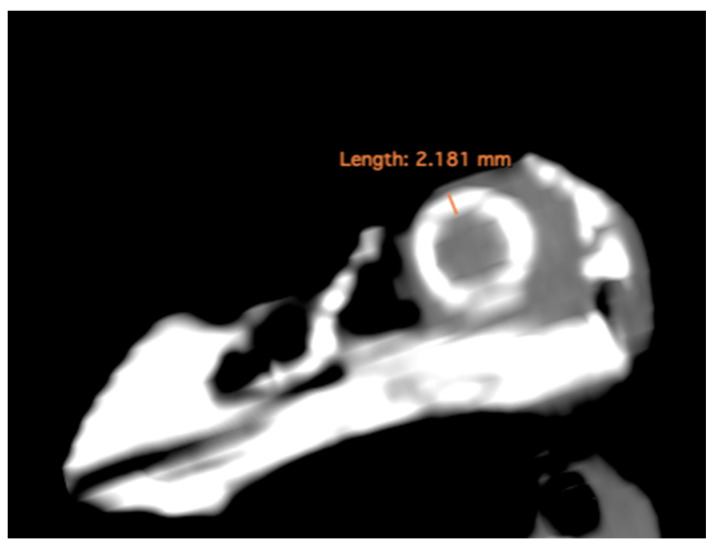
MPR related to the ocular bulb of *F. Arctica* with a measure of the sclerotic ring thickness.

**Figure 3 animals-13-02418-f003:**
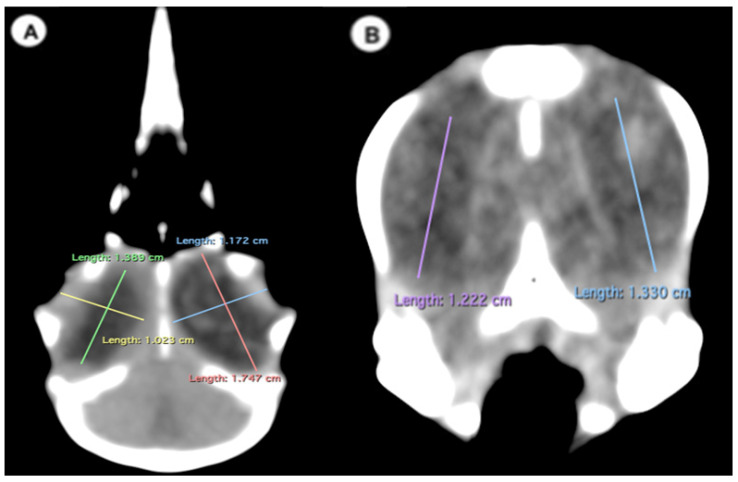
Dorsal multiplanar reconstruction (MPR) and transverse CT images in soft tissue windows of the head with measurements of the length, width (**A**), and height (**B**) of the ocular bulb of the *F. Arctica* head.

**Figure 4 animals-13-02418-f004:**
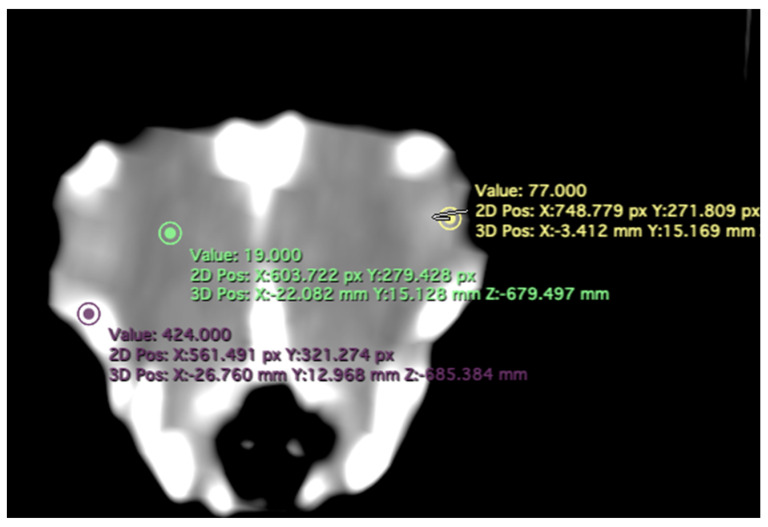
Transverse CT image in the soft tissue window of the *F. Arctica* head. Regions of interest (ROI) indicate the areas used to measure mean attenuation in Hounsfield Units (HU) of the lens (yellow circle), sclerotic ring (purple circle), and vitreous humor (green circle).

**Figure 5 animals-13-02418-f005:**
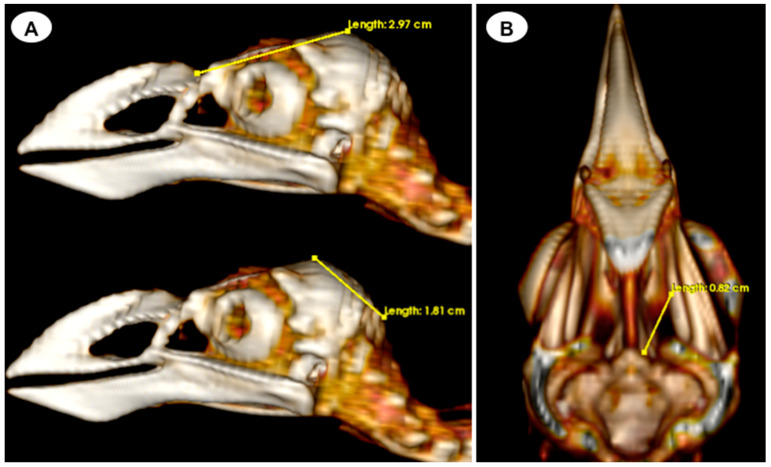
Volume rendering image of the *F. Arctica* skull with measurements of head length (**A**) and orbit depth (**B**). Image (**B**) has been modified by removing the cranial vault in order to measure the depth of the orbital basin.

**Table 1 animals-13-02418-t001:** Measurements of the right and left eyes of the Atlantic puffin.

	Right Eye				Left Eye				Both Eyes			
	Mean	Median	Range	SD	Mean	Median	Range	SD	Mean	Median	Range	SD
Lens diameter (mm)	3.34	3.3	2.7–4.2	0.36	3.30	3.2	2.8–3.9	0.29	3.32	3.3	2.7–4.2	0.32
Internal diameter of the sclerotic ring (cm)	0.64	0.63	0.59–0.69	0.03	0.63	0.63	0.59–0.7	0.03	0.63	0.63	0.59–0.7	0.03
External diameter of the sclerotic ring (cm)	1.44	1.44	1.38–1.48	0.03	1.43	1.43	1.38–1.49	0.03	1.44	1.44	1.38–1.49	0.03
Sclerotic ring thickness (mm)	2.15	2.14	2.04–2.28	0.07	2.18	2.195	2.08–2.29	0.06	2.16	2.16	2.04–2.29	0.06
Ocular bulb height (cm)	1.54	1.52	1.47–1.61	0.04	1.52	1.52	1.41–1.61	0.05	1.53	1.52	1.41–1.61	0.05
Ocular bulb width (cm)	1.29	1.29	1.1–1.42	0.09	1.27	1.28	1.1–1.39	0.08	1.28	1.29	1.1–1.42	0.08
Ocular bulb length (cm)	1.62	1.64	1.46–1.7	0.07	1.60	1.6	1.48–1.71	0.07	1.61	1.62	1.46–1.71	0.07
Orbit depth (mm)	8.16	8.2	7.9–8.4	0.16	8.23	8.3	7.9–8.5	0.18	8.19	8.2	7.9–8.5	0.18
Lens attenuation (HU)	70.14	71	64–79	2.95	68.93	69	64–74	2.72	69.53	69.5	64–79	2.88
Vitreous humor attenuation (HU)	35.17	35	31–41	3.06	35.59	36	31–45	3.76	35.38	35.5	31–45	3.40
Sclerotic ring attenuation (dorsal arch) (HU)	721.21	725	687–752	18.39	719.79	723	687–745	17.64	720.50	723	687–752	17.87

## Data Availability

The information is available at “https://accedacris.ulpgc.es”, accessed on 25 June 2023.

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
