# Peer review of "A Cadaveric Study Using Computed Tomography for Measuring the Ocular Bulb and Scleral Skeleton of the Atlantic Puffin (Aves, *Alcidae*, *Fratercula arctica*)"

_animals, 2023, doi:10.3390/ani13152418_

Round 1
Reviewer 1 Report
This manuscript includes the results of a computed microtomography study on the eye of. The technique makes it possible to obtain information on aspects that are very difficult to analyze using standard techniques. Some images do not have very good quality, but the authors correctly justify the causes. The results are very interesting and original However, the discussion needs a rewrite. In this section the authors limit themselves to presenting the advantages of this technique and mentioning in which groups of animals it was used. It would be necessary to discuss the results in a physiological and phylogenetic framework that relates the results with what is known in other birds and to speculate about what differences could be related to the aquatic habitat of this species.
Author Response
Dear Reviewer,
Following your recommendation, we have included additional information in the discussion section, focusing on the physiological and phylogenetic aspects that relate the results with what is known in other birds and speculate about what differences could be related to the aquatic habitat of this species.
Reviewer 2 Report
Thank you for your invitation to review the manuscript titled “Computed tomography measurements of the Eye-ball and Scleral Skeleton of the Atlantic Puffin (Fratercula arctica)”. This is an interesting cadaveric and morphometric study performed in ocular bulbs of one avian species from computed tomography. However, some major and minor revisions should be performed on the manuscript.
Major revisions
The title should specify that it is a cadaveric study since the measurements could be different in live animals.
One of the conclusions of this study was a comprehensive understanding of the anatomy of the eyeball and the sclerotic ring in the Atlantic puffin. However, I do not agree with the first two lines of the conclusion “the CT images obtained in this study provided a comprehensive understanding of the anatomy of the eyeball and the sclerotic ring in the Atlantic puffin” since this is only a morphometric study of the ocular bulb in one avian species from computed tomography. The study would be complete if the authors include specific dissections of the ocular bulb. Besides, the authors should include an evolutionary relationship in the discussion section. For example, if the measurements of the ocular bulb and scleral ring have a relationship with its lifestyles.
Minor revisions
In the title, please include the taxonomic family of the species and the scientific name should be in italics.
In introduction, all scientific names should be written with Italics.
Change the words “globe” and “eyeball” by ocular bulb (bulbus oculi).
In materials and methods, include the use of the terminology of Nomina Anatomica Avium.
Other minor revisions
Lines
36: change the word “back” by “dorsum”
37: change the word “belly” by “ventrum”
161: change the words globe and eyeball by ocular bulb.
169: Is it foramen or canal?
186-187: cranial pole? Is it “anterior pole”? the anterior pole is the anterior extreme of the cornea. The term cranial is not used in the ocular bulb. However, the scleral ring is not at the cranial pole, therefore, I suggest clarifying this.
252: Our study does not compare…
Lines 284-302: the last two paragraphs are repeated.
The first reference should be cited based on the web page. The IUCN web page has the citation form, please review it.
Is the second reference reliable? I recommend to include another reference.
Author Response
Dear Reviewer,
We appreciate your comments and suggestions since they helped improve our manuscript.
Major revisions
The title has been modified, specifying on it that it is a cadaveric study.
Since this study was focused on morphometric parameters we did not perform ocular dissections. However, it would be quite helpful to include them in further studies on live animals. Moreover, following your recommendations, we have added additional information in the discussion section about ocular measurements and activity patterns.
Minor revisions
In the title, we have included the taxonomic family of the species and the scientific name in italics.
In the introduction, all scientific names have been rewritten with Italics.
In the materials and methods section, we have included the use of the terminology of Nomina Anatomica Avium.
Minor comments
Line 36: As you recommend, we have changed the word “back” to “dorsum”.
Line 37: Following your suggestion, we have changed the word “belly” to “ventrum”.
Line 161: We have changed the words globe and eyeball by ocular bulb along the manuscript.
Line 169: Is it foramen or canal? It is foramen, folowing the Nomina Anatomica Avium.
Line 186-187: cranial pole? Is it “anterior pole”? Following your suggestion, we have clarified and corrected this statement replacing "cranial" with "anterior"
Line 252: We have added your suggestion "Our study does not compare…"
Lines 284-302: We have deleted one of the two paragraphs, and the other has been moved to the conclusions section.
As you recommend, we have done the changes in the references. However, the second one is an online database page created by Michigan State University, and quite used by veterinarians or biologists that work with wildlife species.
Reviewer 3 Report
Authors should review the applied statistical treatments. Mainly the application of the ANOVA and also the correlation test. Still regarding the correlation test, authors should consult related literature. Thus, aiming at the interpretation of the obtained values and their intensities. In this way, the work should be reformulated and its results properly interpreted.

Authors should review the applied statistical treatments. Mainly the application of the ANOVA and also the correlation test. Still regarding the correlation test, authors should consult related literature. Thus, aiming at the interpretation of the obtained values and their intensities. In this way, the work should be reformulated and its results properly interpreted.
Author Response
Dear Reviewer 3
We appreciate your comments and suggestions. Therefore, the statistical analysis has been reviewed and changed following your advice.
The material and methods have been rephrased as:
“The statistical analysis was performed using commercially available software (SPSS 19, Statistical Package for the Social Sciences, Chicago, IL, USA). Descriptive statistics, including the mean, median, range, and standard deviation (SD), were calculated for each measurement. The Shapiro-Wilk test was used to assess the normal distribution of quantitative data. The Mann-Whitney U test was employed to compare measurements between the right and left eyes. Furthermore, Spearman correlation was utilized to analyse the relationship between different ocular bulb variables and head length, as well as other quantitative variables. The statistical significance level was set at p < 0.05.”
And the results:
Table 1 presents the measurements of internal structures for the left, right, and both eyes of all 29 puffins included in the study. The mean, median, and standard deviation of the head length were 5.10 cm, 5.07 cm, and 0.13 cm, respectively, with a range of 4.8 to 5.32 cm. The average lens diameter for all eyes was 3.32 mm, with a range of 2.7 to 4.2 mm. The average internal diameter of the sclerotic ring was 0.63 cm, ranging from 0.59 to 0.7 cm. Additionally, the average external diameter of the sclerotic ring was 1.44 cm, ranging from 1.38 to 1.49 cm. The sclerotic ring thickness had a mean value of 2.16 mm, with a range of 2.04 to 2.29 mm. Regarding the eyeballs, they had an average height of 1.53 cm (range: 1.41-1.61 cm), an average width of 1.28 cm (range: 1.1-1.42 cm), and an average length of 1.61 cm (range: 1.46-1.71 cm). Additionally, the orbit depth had a mean of 8.19 mm (range: 7.9-8.5). Some variables exhibited a non-normal distribution. The Mann-Whitney U test indicated no significant differences between the measurements of the right and left eyes. Likewise, the same analysis demonstrated no statistically significant differences between the right and left eyes when considering all variables collectively.
The Spearman correlation analysis conducted to examine the relationship between eye measurements and head length did not reveal any statistically significant findings. However, significant correlations were observed in other aspects. There was a strong correlation between eyeball height and width (rho=0.661; p<0.001), indicating a substantial relationship between these variables. Additionally, moderate correlations were observed between eyeball height and length (rho=0.403; p=0.002), as well as between eyeball width and length (rho=0.449; p=0.002), suggesting meaningful associations between these measurements. Moreover, a good correlation was found between the external and internal diameter of the sclerotic ring (rho=0.431; p<0.001), indicating a relationship between these variables.
Correlations involving the internal diameter of the sclerotic ring showed statistical significance with eyeball height (rho=0.601; p<0.001), length (rho=0.496; p<0.001), but not with the width (rho=0.203; p=0.127). In a similar way, the external diameter of the sclerotic ring exhibited
significant correlations with eyeball height (rho=0.688; p<0.001) and width (rho=0.642; p<0.001), while in this case no significant correlation was found with eyeball length (rho=-0.059; p=0.662). Significant correlations were observed when comparing the orbit depth with eyeball measurements: height (rho=-0.484; p<0.001) and width (rho=-0.294; p=0.025). However, no significant correlation was observed with eyeball length (rho=-0.114; p=0.395).
Round 2
Reviewer 1 Report
The authors realizad changes and the manuscript could be published
Author Response
Dear Reviewer,
We really appreciate your comments and recommendations, which have been quite helpful in improving our manuscript
Sincerely
Reviewer 2 Report
This is an improved version of the former article. It is ready for publication after some minor corrections:
I recommend this title: A Cadaveric Study using Computed Tomography for Measuring the Ocular Bulb and Scleral Skeleton of the Atlantic Puffin (Aves, Alcidae, Fratercula arctica).
Lines 35-36: Please include words different from the title to improve the finding this article in the internet browsers.
It is unnecessary to include the scientific name between parenthesis again after the first mention: Lines 105, 138, and 371.
Lines 283-285: The anterior pole is the anterior extreme of the cornea. Therefore, a better redaction would be: …the sclerotic rings are visible from the anterior view of the respective eyes…
Line 431: I suggest to change the term “non-extinct” by “extant”.
Author Response
Dear Reviewer,
We really appreciate your comments and recommendations, which have been immensely helpful in improving our manuscript
- I recommend this title: A Cadaveric Study Using Computed Tomography for Measuring the Ocular Bulb and Scleral Skeleton of the Atlantic Puffin (Aves, Alcidae, Fratercula arctica). As you recommend, we have modified the title.
- Lines 35-36: Please include words different from the title to improve this article's finding in the internet browsers. Following your suggestion, we have included new keywords in the list.
- It is unnecessary to include the scientific name between parenthesis again after the first mention: Lines 105, 138, and 371. As you suggested, we have corrected this problem.
- Lines 283-285: The anterior pole is the anterior extreme of the cornea. Therefore, a better redaction would be: …the sclerotic rings are visible from the anterior view of the respective eyes… We have changed the redaction as you suggested.
- Line 431: I suggest to change the term “non-extinct” by “extant”. We have replaced non-extinct by extant.
Reviewer 3 Report
accept
accept
Author Response
Dear Reviewer,
We really appreciate your comments and suggestions, which have been quite helpful in improving the quality of our manuscript
Sincerely